# The Nuclear NF-κB Regulator IκBζ: Updates on Its Molecular Functions and Pathophysiological Roles

**DOI:** 10.3390/cells13171467

**Published:** 2024-08-31

**Authors:** Soh Yamazaki

**Affiliations:** Department of Biochemistry, Toho University School of Medicine, 5-21-16 Omorinishi, Ota-ku, Tokyo 143-8540, Japan; syamaz@med.toho-u.ac.jp; Tel.: +81-3-3762-4151; Fax: +81-3-5493-5412

**Keywords:** IκB, NF-κB, transcriptional regulation, IL-17, epithelial cells, POU

## Abstract

More than a decade after the discovery of the classical cytoplasmic IκB proteins, IκBζ was identified as an additional member of the IκB family. Unlike cytoplasmic IκB proteins, IκBζ has distinct features, including its nuclear localization, preferential binding to NF-κB subunits, unique expression properties, and specialized role in NF-κB regulation. While the activation of NF-κB is primarily controlled by cytoplasmic IκB members at the level of nuclear entry, IκBζ provides an additional layer of NF-κB regulation in the nucleus, enabling selective gene activation. Human genome-wide association studies (GWAS) and gene knockout experiments in mice have elucidated the physiological and pathological roles of IκBζ. Despite the initial focus to its role in activated macrophages, IκBζ has since been recognized as a key player in the IL-17-triggered production of immune molecules in epithelial cells, which has garnered significant clinical interest. Recent research has also unveiled a novel molecular function of IκBζ, linking NF-κB and the POU transcription factors through its N-terminal region, whose role had remained elusive for many years.

## 1. Introduction

The transcription factor NF-κB plays a central role in immune responses by regulating the production of immunity-associated molecules, such as antimicrobial proteins and pro-inflammation mediators [1,2]. In resting cells, NF-κB is prevented from entering the nucleus, because it is sequestered by cytoplasmic IκB proteins. The interaction between NF-κB and IκB is mediated by the Rel homology domains of NF-κB and the ankyrin repeat domains of IκB [1,2]. Upon stimulation by agents related to infection and inflammation, such as Toll-like receptor (TLR) ligands, TNF, or IL-1β, the signaling from these receptors leads to the ubiquitin-dependent degradation of cytoplasmic IκB proteins. The liberated NF-κB enters the nucleus, where it participates in the transcriptional activation of κB site-harboring target genes.

IκBζ was identified as a protein containing ankyrin repeats, but it regulates NF-κB in a manner distinct from cytoplasmic IκB proteins. IκBζ exerts its function after NF-κB has entered the nucleus, enabling NF-κB to selectively regulate a specific set of genes. Although the exact mechanism of how IκBζ regulates NF-κB is not yet fully understood, recent findings regarding the first structural basis of IκBζ and the genome-wide analysis of IκBζ occupancy have significantly advanced our understanding. Additionally, the previously unclear role of the N-terminal half of IκBζ has been elucidated by identifying a consensus amino acid sequence that interacts with POU transcription factors, including OCT family members. This sequence was functional in the cooperative transcriptional regulation by NF-κB and OCT2. In terms of the in vivo function of IκBζ, its role in IL-17-triggered gene regulation is crucial for the maintenance of epithelial cell homeostasis. Studies on human diseases and knockout mice have demonstrated that IκBζ-mediated gene regulation not only plays a protective role in host defense but also contributes to the pathogenesis of certain diseases. Based on these findings, there have been trials targeting the IL-17–IκBζ pathway for therapeutic purposes, overcoming the difficulty associated with the intracellular delivery of drugs.

## 2. Identification, Structure, and Expression Properties

The gene encoding IκBζ, known as *Nfkbiz* in mice, was cloned through the screening of genes upregulated in response to stimulation with lipopolysaccharide (LPS) or IL-1β [3,4,5]. IκBζ contains an ankyrin repeat domain, which is characteristic of the IκB family and is responsible for its interaction with NF-κB (Figure 1) [1,2]. Unlike cytoplasmic IκB family members, IκBζ and other nuclear IκB members preferentially bind to the NF-κB p50 subunit, rather than p65 and c-Rel, in the nucleus (Table 1) [5,6,7,8,9,10,11]. While the amino acid sequences of the overall ankyrin repeats are similar among the nuclear IκB subfamily members, a specific amino acid residue (Asp-451 in murine IκBζ) was found to be essential for binding to the p50 subunit [12]. This residue is completely conserved among nuclear IκB members but is replaced by a different amino acid in cytoplasmic IκB members (Figure 1). The significance of the shared preference for p50 among nuclear IκB proteins is unclear, while cytoplasmic IκB proteins are thought to bind to p65 and c-Rel for preventing the unwanted transcriptional initiation in resting cells. Given the absence of the transactivation domain (TAD) in p50, it is conceivable that a complex formed by a nuclear IκB and a p50 homodimer could play a role in specific biological contexts [7,8], although this is unlikely in the case of IκBζ (see below). In addition, while cytoplasmic IκB proteins are degraded in response to stimuli, signal-induced proteolysis has not been known for nuclear IκB members.

The basal expression of IκBζ in resting cells is very low, which contrasts with the consistent expression levels of cytoplasmic IκB proteins for preventing the nuclear entry of NF-κB [1,2]. The expression of *Nfkbiz* is highly induced following stimulation with TLR ligands such as LPS, and IL-1β, via transcriptional activation by NF-κB (Figure 2) [5,13,14]. Intriguingly, TNF strongly activates NF-κB but only minimally induces the expression of *Nfkbiz* [4,5]. This is because TNF receptor signaling fails to lead to the stabilization of *Nfkbiz* mRNA [14]. MyD88 is an adaptor molecule that is employed by the signaling from TLRs or the IL-1 receptor but not from the TNF receptor [15]. In MyD88-deficient cells, LPS still can activate NF-κB [16] but fails to induce *Nfkbiz* expression [10,17], suggesting that MyD88 transmits the specific mRNA stabilization signal from TLRs and the IL-1 receptor. In addition to TLR ligands and IL-1β, the other pro-inflammatory cytokine IL-17 can also contribute to IκBζ induction (Figure 2) [14,18,19]. The signaling from the IL-17 receptor involves the adaptor protein Act1, which shares a weak sequence similarity with MyD88 [20] and activates the common downstream molecule TRAF6 (Figure 2) [21,22]. Consistent with the finding that IL-17 only weakly activates NF-κB while enhancing the stability of *Nfkbiz* mRNA [19], co-stimulation with TNF and IL-17 strongly induces *Nfkbiz* expression [14]. While the transcription of *Nfkbiz* is primarily regulated by NF-κB, the contribution of other transcription factors, such as STAT3, has also been reported [23,24,25]. The post-transcriptional regulation of IκBζ expression occurs at both the mRNA stability and translation efficiency levels [26,27]. 

The endoribonuclease Regnase-1/MCPIP1 (encoded by *Zc3h12a*) plays a crucial role in the post-transcriptional regulation via a stem-loop structure in the 3′ region of *Nfkbiz* mRNA [28,29,30,31,32]. Additionally, the RNA-binding protein Arid5a counteracts the Regnase-1-mediated downregulation by binding to the 3′-untranslated region of *Nfkbiz* mRNA [27]. 

Similar to *Nfkbiz*, the expression of *Zc3h12a* and *Arid5a* is upregulated following IL-17 stimulation [27,33,34], indicating a tight control by multiple factors. Furthermore, microRNAs (miRNAs) have been reported to be involved in the post-transcriptional regulation of *Nfkbiz* mRNA [35,36,37,38,39]. The amount of IκBζ protein can be modulated by the electrophilic agent itaconate or its derivatives independently of the antioxidant transcription factor NRF2 [40,41]. Itaconate inhibited the LPS-stimulated increase in the IκBζ protein without affecting the NF-κB activation or *Nfkbiz* mRNA levels, and co-treatment with the antioxidant *N*-acetylcysteine (NAC) or reduced glutathione (GSH) reverses the inhibitory effect of itaconate [40]. On the other hand, it was reported that NAC abrogates the IκBζ protein expression induced by co-stimulation with TNF and IL-17 in colon epithelial cells [42]. Similarly, reactive oxygen species (ROS) have been reported to mediate the IκBζ protein upregulation in IL-1β-stimulated chondrocytes [43]. Consistent with the observation that ROS was generated following IL-1β stimulation via the lactate dehydrogenase A (LDHA)-mediated promotion of the electron donor activity of NADH, the abrogation of LDHA by gene deletion or its inhibitor reduced the expression of IκBζ protein [43]. The proteasomal degradation of IκBζ protein was reported to be facilitated by the NAD(P)H:quinone oxidoreductase 1 (NQO1)-mediated association of IκBζ with the E3 ligase PDLIM2 in LPS-activated macrophages [44], although it remains unclear whether the antioxidant enzymatic activity of NQO1 is necessary or not. The NQO1-mediated facilitation of IκBζ degradation was also observed in IL-20-stimulated hepatocytes [45].

## 3. Molecular Functions

The initial characterization of the function of IκBζ using κB site-containing reporters yielded controversial results regarding whether IκBζ serves as a positive or negative regulator of NF-κB [3,5,46,47]. However, analyses of the cells derived from *Nfkbiz*-deficient mice have clarified that IκBζ is essential for the transcriptional activation of a subset of NF-κB target genes (Figure 3) [10]. While the specific molecular features of target genes that define IκBζ dependence have not been fully elucidated, significant insights have been gained from the investigations on *Lcn2*, a typical IκBζ-dependent gene that encodes the antimicrobial protein Lipocalin-2, as known as neutrophil gelatinase-associated lipocalin (NGAL) [48,49,50,51]. Chromatin immunoprecipitation (ChIP) experiments demonstrated that LPS-stimulated recruitment of the transcription activators, including NF-κB p65, c-Rel, C/EBPβ, RNA polymerase II, TATA box-binding protein (TBP), to the *Lcn2* promoter is all negated in *Nfkbiz*-deficient macrophages [52,53]. The recruitment of BRG1, a subunit of the chromatin-remodeling SWI/SNF complex, is also abolished in the absence of IκBζ, indicating that IκBζ increases the accessibility of the *Lcn2* promoter by altering the chromatin structure [53]. Similarly, IκBζ-mediated chromatin remodeling at the *Il6* promoter has been reported [54]. In wild-type macrophages, NF-κB p65 and other transcription activators are recruited more rapidly to the promoter of the IκBζ-independent gene *Cxcl2* compared to the *Lcn2* promoter following LPS stimulation. The p65 recruitment to the *Cxcl2* promoter is not affected by the absence of IκBζ [52,53]. It has been known that while the p65-containing NF-κB dimers translocate to the nucleus immediately after LPS stimulation, the recruitment to certain promoters is delayed due to the time required for the stimulus-induced chromatin modification and acquisition of promoter accessibility [55]. IκBζ is considered to be produced as a product of a primary response gene, whose mRNA expression is induced independently of de novo protein synthesis (Figure 3) [53,56,57] and subsequently participates in the activation of secondary response genes such as *Lcn2* and *Il6*. Besides the property of the chromatin structure of the *Lcn2* locus, detailed investigations into the *Lcn2* promoter sequence have identified a pyrimidine-rich motif located downstream of the κB site, which is required for binding by the IκBζ–NF-κB complex [12,51].

Consistent with the preferential binding of IκBζ to the p50 subunit, a functional link between IκBζ and p50 has been observed in the transcriptional activation of target genes [10,58,59]. In cells lacking *Nfkb1* (encoding p50), IκBζ is unable to induce the expression of *Il6* and *Lcn2* [10,58,59]. The association of IκBζ with the *Lcn2* promoter following LPS stimulation is entirely dependent on the p50 subunit [58]. Genome-wide transcriptome analyses using LPS-stimulated macrophages deficient in *Nfkbiz* or *Nfkb1* uncovered a considerable overlap between IκBζ-dependent genes and p50-dependent genes [59]. Importantly, the ChIP-seq analysis elucidated that the dependence on IκBζ and p50 cannot be explained by their selective targeting to target genes [59]. For instance, IκBζ is recruited to the *Cxcl2* promoter after LPS stimulation, despite being dispensable for *Cxcl2* expression [58]. In contrast to the situation at the *Lcn2* promoter, where p65 association requires IκBζ, ATAC-seq and ChIP-seq analyses indicated that most IκBζ-binding loci are already accessible before stimulation and are pre-bound by p65–p50 heterodimers after LPS stimulation [59]. Furthermore, it appears that p65–p50 heterodimers, rather than p50 homodimers, play a major role in IκBζ-mediated gene regulation, as the majority of the IκBζ peaks in ChIP-seq coincide with genomic loci that are co-bound by p65 and p50, but not those bound by p50 alone [59]. Zhu et al. recently reported the X-ray crystal structure of IκBζ in a complex with the p50 homodimer and showed that the nuclear localization signal (NLS)-containing polypeptide from one of the two p50 subunits was sufficient for the binding of the dimer to IκBζ [60]. It was also demonstrated that, similar to p50 homodimers, the p65–p50 heterodimer can form a complex with IκBζ on κB DNA [60]. This complex formation on the DNA is well compatible with the findings in the abovementioned ChIP-seq data, which show genomic occupancy by the p65–p50 heterodimer with IκBζ [59].

While the function of the N-terminal region of IκBζ remained elusive for a long time, Alpsoy et al. identified a short stretch that is conserved among coactivator proteins for POU transcription factors (Figure 1) [61]. IκBζ indeed directly bound to the POU family members, such as OCT2 and OCT6, and activated promoters containing both κB and octamer sites [61]. ChIP-seq using lymphoma cells (in which IκBζ expression is constitutively high because of the gain-of function mutation of *MYD88*) revealed a significant overlap in genomic occupancy by IκBζ, OCT2, and NF-κB p50 [61]. Furthermore, RNA-seq analyses uncovered an overlap between genes downregulated by the knockout of IκBζ and OCT2, confirming the functional correlation in vivo [61]. An IκBζ-mediated collaboration between NF-κB and OCT2 was also observed in the transcriptional activation of *Ifng* (encoding IFN-γ) in NK cells [62]. The expression of *Nfkbiz* and *Pou2f2* (encoding OCT2) was upregulated in activated NK cells [62].

## 4. Cell Type-Specific Roles of IκBζ and Associations with Human Diseases

### 4.1. Keratinocytes

The Germ-line deletion of *Nfkbiz* in mice results in atopic dermatitis-like skin lesions with acanthosis and lichenoid tissue change in the periocular epithelium 4 to 5 weeks after birth [10,63]. These mutant mice also developed dacryoadenitis, which is similar to human Sjögren’s syndrome, an autoimmune disease [23]. A massive infiltration of lymphocytes was observed in the lesional tissues of *Nfkbiz*^−/−^ mice, and the absence of lymphocytes alleviated the disease as seen in *Nfkbiz*^−/−^; *Rag2*^−/−^ mice [23]. Although this raised the possibility of IκBζ-deficient lymphocytes playing a pathogenic role, the deletion of *Nfkbiz* in T cells or B cells did not cause the development of the disease [23]. A bone marrow transfer experiment supported the conclusion that non-hematopoietic cells are responsible for the pathogenesis. The pathologies observed in *Nfkbiz*^−/−^ mice were replicated in keratinocyte-specific *Nfkbiz*-deficient (*Nkfbiz*^fl/fl^; *KRT5-Cre*) mice (Table 2) [23]. A subsequent microbiome analysis of the skin tissue of the *Nkfbiz*^−/−^ mice revealed a reduced diversity of commensal bacteria with a marked expansion of *Staphylococcus xylosus*. Importantly, the administration of antibiotics ameliorates the dermatitis of *Nkfbiz*^−/−^ mice [64]. Furthermore, the epicutaneous application of *Staphylococcus aureus* led to a higher skin burden of the bacteria in *Nfkbiz*^fl/fl^; *KRT5-Cre* mice, due to a defect in the production of anti-microbial proteins in the skin (Table 2) [65]. Thus, the dermatitis in *Nfkbiz*-deficient mice appears to be caused by a defect in controlling skin bacteria.

Conversely, a higher expression of IκBζ increases the risk of another skin disease, psoriasis (Figure 4). Tsoi et al. identified *NFKBIZ* as a psoriasis susceptibility locus in humans by a meta-analysis of multiple GWAS data sets, with its expression being elevated in the lesional psoriatic skin of affected patients [66]. Consistent with the fact that IL-17 has a pathogenic role in psoriasis by inducing pro-inflammatory molecules [67], treatment with an anti-IL-17A antibody, such as secukinumab, has shown sustained efficacy in psoriatic arthritis [68,69,70]. It was found that IκBζ mediates the IL-17-induced expression of psoriasis-associated genes in the skin, and *Nkfbiz*^−/−^ mice are resistant to an imiquimod-induced psoriasis-like skin inflammation model [71]. The local abrogation of IκBζ by the intradermal injection of siRNA or specific gene deletion using the *KRT14*-Cre driver also rendered mice resistant to the psoriasis model (Table 2) [71,72]. The other cytokine IL-36 is also associated with psoriasis, particularly in the generalized pustular psoriasis subtype, and IκBζ was shown to mediate the IL-36-induced development of the psoriasis model in mice [73]. Notably, the topical delivery of *Nfkbiz* siRNA into the skin with the aid of ionic liquids successfully suppressed the imiquimod-induced psoriasis model in mice [74]. Similarly, it was demonstrated that the downregulation of IκBζ by the administration of dimethyl itaconate, a membrane permeable derivative of itaconate, allowed mice to be resistant to the psoriasis model [40].

### 4.2. Intestinal Epithelial Cells

The whole-exome sequencing of colon epithelial cells from patients with ulcerative colitis (UC), an inflammatory bowel disease (IBD), conducted by two research groups, unveiled the accumulation of somatic mutations in several genes related to IL-17R signaling [75,76]. Notably, loss-of-function mutations in *NFKBIZ* were most frequently identified in these studies. Although the role of *NFKBIZ* mutations in the pathogenesis of UC remains unclear, it appears that the defect in IL-17R signaling due to the *NFKBIZ* mutations may provide a survival advantage under UC conditions, where epithelial cells are continuously exposed to IL-17. Indeed, it was demonstrated that the inhibition of IL-17R signaling or the genetic deletion of *NFKBIZ* allowed for colon epithelial organoids to counteract the toxic effects of IL-17 under certain culture conditions [75]. A mixed co-culture experiment revealed that *Nfkbiz*-deficient cells have a selective advantage over *Nfkbiz*-sufficient cells when cultured in the presence of IL-17A. In a mouse model, *Nfkbiz*-deficient epithelial cells became to occupy a larger area of the colon compared to *Nfkbiz*-sufficient cells following the induction of colitis with dextran sulfate sodium (DSS) [76].

The deletion of *Nfkbiz* in intestinal epithelial cells using the *Vil1-Cre* driver leads to dysbiosis in the small intestine due to the failure of the IL-17-stimulated production of anti-microbial molecules (Table 2) [58]. A distinctive feature of this dysbiosis was the expansion of segmented filamentous bacteria (SFB), which are known to facilitate the development of Th17 [77]. Consistently, Th17 cells were markedly increased, and the expression of Th17-associated cytokine genes, including *Il17a*, *Il17f*, and *Il22*, was elevated in the small intestine of *Nfkbiz*^fl/fl^*Vil1-Cre* mice [58]. Despite a large amount of IL-17 being released from the expanded Th17 cells in the intestine of the mutant mice, the epithelial cells fail to express antimicrobial genes in response to IL-17 due to the lack of IκBζ. Similar phenotypes were observed in mice lacking IL-17RA specifically in intestinal epithelial cells [78]. These findings align with the observation that clinical trials targeting IL-17 signaling for the treatment of Crohn’s disease often resulted in rather aggravation of the disease, potentially due to inducing dysbiosis [79,80].

### 4.3. Oral Epithelial Cells

The importance of IκBζ in IL-17-stimulated gene activation was demonstrated in oral epithelial cells during oropharyngeal candidiasis [81]. Following oral infection with *Candida albicans*, the expression of IκBζ was increased in the tongues of wild-type mice, a response that was abolished by the deletion of IL-17RA [81]. The systemic deletion of *Nfkbiz* (*Nfkbiz*^fl/fl^*Rosa26*^CreERT2^), as well as deletion specifically in the suprabasal epithelial layer (SEL, *Nfkbiz*^fl/fl^*Krt13-Cre*), led to an increase in fungal loads after oral infection with *C. albicans* (Table 2) [81]. Among the genes upregulated via the IL-17R–IκBζ axis during fungal infection, *Defb3* (encoding β-defensin 3) was found to be important for immunity against the fungal infection.

### 4.4. Natural Killer (NK) Cells

IκBζ has also been shown to play a significant role in NK cells. Miyake et al. showed that IκBζ is expressed in NK cells upon stimulation with IL-12 and IL-18 and is essential for the activation of NK cells, which includes the production of IFN-γ and cytotoxicity against target cells [82]. *Nfkbiz*^−/−^ mice exhibited higher mortality when infected with mouse cytomegalovirus (MCMV), a virus typically eliminated by NK cells [82]. The IκBζ-dependent production of IFN-γ was also observed in human CD56^+^ NK cells following stimulation with IL-12 and IL-18 [83]. It was reported that there is an involvement of IκBζ in the IFN-γ production of KG-1 cells, a human acute myeloid leukemia cell line, after simultaneous stimulation with IL-1β, IL-18, and TNF [84]. Additionally, it was shown that the deletion of Regnase-1/MCPIP1 in NK cells leads to the upregulation of IκBζ, resulting in an enhanced production of IFN-γ [62].

### 4.5. Chondrocytes

Osteoarthritis is a degenerative joint disease characterized by the destruction of joint cartilage. It was found that the *Nfkbiz* expression is elevated in chondrocytes upon stimulation associated with osteoarthritis, such as the surgical destabilization of the medial meniscus in mice [43,85]. The chondrocyte-specific deletion of *Nfkbiz* using the *Col2a1-Cre* driver in mice led to the downregulation of matrix-degrading enzymes and the alleviation of osteoarthritis after the surgery (Table 2) [85]. The IL-1β-induced expression of genes encoding the matrix metalloproteinases (MMP-3 and MMP-13) and the aggrecanase ADAMTS-5 was reduced in primary chondrocytes isolated from conditional *Nfkbiz*-deficient mice [85]. Additionally, it was shown that the ROS-dependent stabilization of IκBζ likely exacerbates experimental osteoarthritis in mice [43].

### 4.6. Other Types of Cells

In acute hepatitis models induced by the injection of concanavalin A or *Klebsiella pneumonia*, IκBζ plays defensive roles by inducing the genes *Il6* and *Lcn2*, encoding the hepatoprotective cytokine and the anti-microbial protein, respectively (Table 2) [45]. The deletion of *Nfkbiz* in hepatocytes using the *Alb-Cre* driver facilitated the progression of nonalcoholic fatty liver disease (NAFLD) in a mouse model that was induced by feeding with a choline-deficient, L-amino acid-defined, high-fat diet (CDAHFD) [86]. IκBζ appears to prevent hepatic steatosis by regulating genes associated with triglyceride metabolism [86].

It was reported that *Nfkbiz*-deficient naive CD4^+^ T cells fail to differentiate into Th17 cells, and *Nfibiz*^−/−^ mice are resistant to experimental autoimmune encephalomyelitis (EAE), a disease where Th17 cells play a major pathogenic role [87]. In contrast to its role in NK cells, IκBζ was found to negatively regulate IFN-γ production in naive CD4^+^ T cells (Table 2) [88]. The deletion of *Nfkbiz* in B cells using the *Cd79a-Cre* (*Mb1-Cre*) driver resulted in the impairment of the T cell-independent type 1 antibody response, which is induced by a 2,4,6-trinitrophenol (TNP)-conjugated LPS (Table 2) [89]. It was shown that IκBζ plays a crucial role in class switch recombinations in B cells by upregulating *Aicda*, which encodes activation-induced cytidine deaminase (AID) [89].

The IL-1 family member IL-33 is a ligand for ST2L (as known as IL-1R4), which is highly expressed in mast cells, Th2 cells, and group 2 innate lymphoid cells (ILC2). Consistent with MyD88 being the adaptor in IL-33R signaling (Figure 2), the stimulation of mast cells with IL-33 induced *Nfkbiz* expression, and the deletion of *Nfkbiz* in mast cells resulted in the reduction in a set of pro-inflammatory genes [90]. Given the potent activity of IL-33 in inducing type 2 cytokines in mast cells, IκBζ might become a potential therapeutic target for allergic diseases.

Alterations of the *NFKBIZ* expression in various human cancer tissues and the potential roles of IκBζ in cancer development have been described, although its role in cancer appears to be more complicated than was previously thought [39,91,92,93,94,95,96,97].

**Table 2 cells-13-01467-t002:** Phenotypes of mice carrying cell-type-specific deletions of *Nfkbiz*.

Tissue (Cre Driver)	Phenotypes
Epithelial cells (*KRT5*-Cre)	Inflammation in facial skin, dacryoadenitis similar to human Sjögren’s syndrome [23], reduced production of antimicrobial proteins, and susceptible to infection by *Staphylococcus aureus* [65].
Keratinocytes (*KRT14*-Cre)	Resistance to imiquimod-induced psoriasis model [72].
Intestinal epithelial cells (*Vil1*-Cre)	Reduced expression of anti-microbial proteins, expansion of SFB, and increase in Th17 cells [58].
Oral epithelial cells (*Krt13*-Cre)	Increased susceptibility to oropharyngeal candidiasis [81].
Chondrocytes (*Col2a1*-Cre)	Downregulation of matrix-degrading enzymes and alleviation of osteoarthritis model [85].
Hepatocytes (*Alb*-Cre)	Facilitated progression of nonalcoholic fatty liver disease (NAFLD) [86].
T cells (*Lck*-Cre)	Increased production of IFN-γ [88].
B cells (*Cd79a/Mb1*-Cre)	Impairment of T cell-dependent type 1 antibody response and Reduction in the expression of activation-induced cytidine deaminase (AID) [89].

## 5. Discussion and Perspectives

While it has been more than twenty years since IκBζ was first identified, our understanding of this molecule continues to evolve. The role of IκBζ in IL-17-triggered gene regulation has garnered significant attention due to its clinical relevance [19,70,98,99,100,101,102,103]. The function of IκBζ in the skin well exemplifies the “double-edged sword” nature of immunity, as IκBζ mediates the production of both antimicrobial proteins and pro-inflammatory mediators (Figure 4) [64,65,98]. While targeting IκBζ might be beneficial for treating psoriasis, the careful management of commensal bacteria and a proper defense against microbial infections are crucial. In intestinal epithelial cells, the deletion of *Nfkbiz* in mice led to dysbiosis and an expansion of pro-inflammatory Th17 cells [58]. Thus, although the absence of IκBζ in epithelial cells might initially suppress the inflammation by lowering pro-inflammatory mediators, the concomitant reduction in antimicrobial activity can indirectly exacerbate inflammation via the dysbiosis-mediated activation of immune cells.

The behavior of IκBζ at a genome-wide scale in LPS-activated macrophages has become clearer [59]. Most genomic loci occupied by IκBζ were found to be already accessible before LPS stimulation and were pre-bound by the p65–p50 heterodimer post-stimulation. This contrasts with previous observations in which IκBζ was thought to play a role in chromatin remodeling to render the target loci accessible to NF-κB p65 [52,53,54]. Although technically difficult, it might be significant to define whether each binding of IκBζ to a certain locus is productive or not. For instance, IκBζ was recruited to the proximal *Cxcl2* promoter [58,59], but the expression of *Cxcl2* does not require IκBζ [52,53,59]. The recruitment of p65 to the *Cxcl2* promoter was rapid and independent of IκBζ, in contrast to the IκBζ-dependent delayed recruitment of p65 to the *Lcn2* promoter [52,53]. If the productive binding of IκBζ to target loci is preceded by the p65–p50 heterodimer, elucidating the unknown role of IκBζ in transcriptional activation is a critical challenge.

IκBζ serves as a transcriptional activator, but its significance lies in preventing target genes from being readily activated by NF-κB (Figure 3). The pathologies in psoriasis caused by aberrantly increased IκBζ well indicate that IκBζ-regulated genes are detrimental if not properly controlled. It is, therefore, reasonable that the expression of IκBζ itself is very strictly regulated at multiple levels. The modulation of IκBζ expression and the mechanisms underlying IκBζ-dependent gene regulation are potential targets for clinical applications aimed at preserving host defenses against infections while mitigating inflammation. Further exploration into the structural determinants of IκBζ-dependent genes, as well as the identification of new IκBζ-interacting partners, could be crucial steps forward.

## Figures and Tables

**Figure 1 cells-13-01467-f001:**
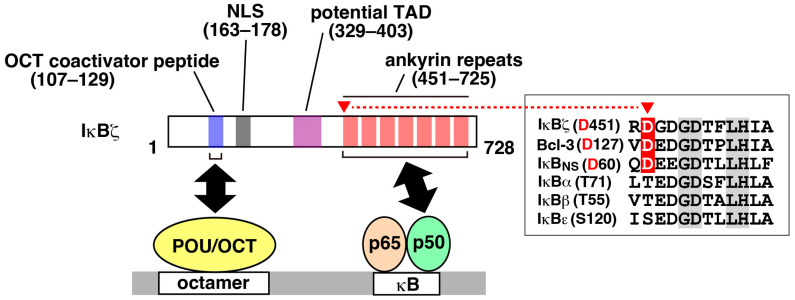
Structure of IκBζ. The numbers for the amino acid residues of mouse IκBζ are presented. Human IκBζ consists of 718 amino acids. IκBζ is likely to activate target genes via interaction with the p65–p50 heterodimer, rather than the p50–p50 homodimer, as indicated by a recent ChIP-seq analysis. IκBζ contains an OCT coactivator peptide responsible for binding to POU transcription factors such as OCT2. IκBζ can binds to both NF-κB and the POU transcription factor to synergistically activate target genes containing κB and octamer sites. Asp-451 in mouse IκBζ, the specific amino acid residue crucial for binding to p50, is indicated by a red triangle. The aspartic acid residue is conserved among nuclear IκB members, as shown in the box on the right (sequences of mouse IκB proteins are presented). This residue of IκBζ is highly conserved across species.

**Figure 2 cells-13-01467-f002:**
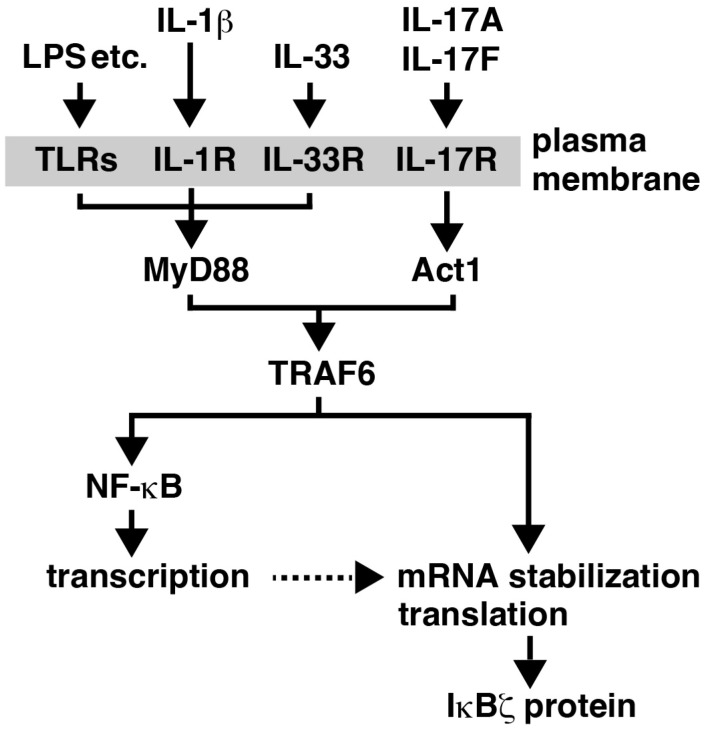
Stimulus-induced expression of IκBζ. The expression of IκBζ is induced upon stimulation with the ligands shown. The IL-33 receptor complex (IL-33R) is composed of ST2L and IL-1RAcP. TNF does not induce the expression of IκBζ, because TNF strongly activates NF-κB but fails to stabilize *Nfkbiz* mRNA [14]. In certain cell types, IL-17 alone poorly induces IκBζ, possibly due to the weak activation of NF-κB; thus, co-stimulation with TNF allows for the strong induction of IκBζ. The amount of IκBζ protein can be influenced by various mechanisms (see text).

**Figure 3 cells-13-01467-f003:**
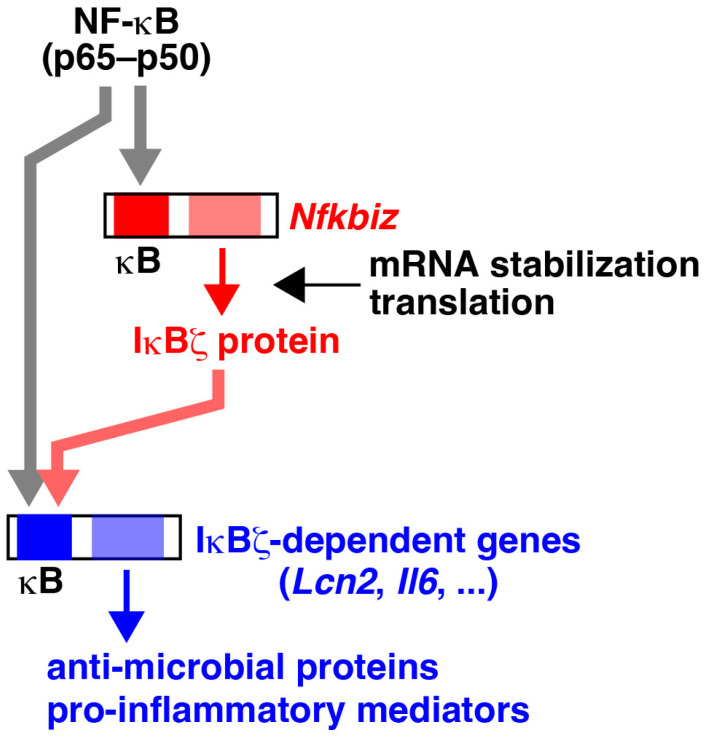
IκBζ-mediated gene regulation. The activated NF-κB (p65–p50 heterodimer) binds to the κB site in the *Nfkbiz* promoter to initiate transcription. *Nfkbiz* is rapidly induced by NF-κB as a primary response gene (shown in red), as its expression is not inhibited by cycloheximide, a protein synthesis inhibitor. The expression of certain NF-κB target genes, such as *Lcn2* and *Il6*, requires both NF-κB and IκBζ. Secondary response genes, including IκBζ-dependent genes (shown in blue), are termed as such because their expression depends on de novo protein synthesis. Since the expression of IκBζ-dependent genes requires an accumulation of IκBζ protein, these genes are selectively induced in response to strong and persistent stimulation. In addition, IκBζ-dependent genes exhibit the same stimulus specificity as the induction of *Nfkbiz*. Thus, NF-κB and IκBζ constitute a coherent feed-forward circuit, allowing for intricate gene regulation. Thick arrows indicate the recruitment of NF-κB (gray) or IκBζ (red) to the κB site of the target genes for transcriptional activation.

**Figure 4 cells-13-01467-f004:**
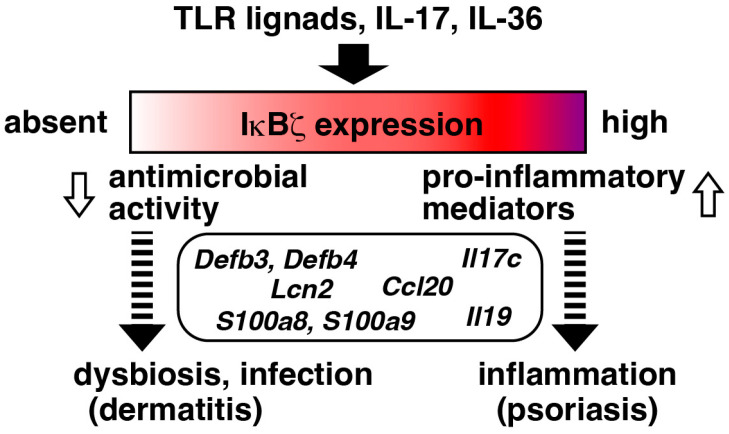
Relationship between IκBζ expression levels and risks of diseases. IκBζ regulates genes associated with antimicrobial proteins and pro-inflammatory mediators. A deficiency or reduced expression of IκBζ may lead to commensal dysbiosis or an increased susceptibility to microbial infections. Conversely, elevated levels of IκBζ can raise the risk of inflammation-related pathologies, such as psoriasis in the skin.

**Table 1 cells-13-01467-t001:** Characteristics of the two IκB subfamilies.

IκB Subfamily	Nuclear IκB	Cytoplasmic IκB
Members	IκBζ, Bcl-3, IκB_NS_	IκBα, IκBβ, IκBε
Intracellular distribution	Nucleus	Cytoplasm
NF-κB regulation	Modulation of NF-κB- mediated transcription	Inhibition of NF-κB nuclear translocation
Preference of binding to NF-κB subunit	p50	p65, c-Rel
Basal expression	Very low	Consistent levels

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
