# Peer review of "The Nuclear NF-κB Regulator IκBζ: Updates on Its Molecular Functions and Pathophysiological Roles"

_cells, 2024, doi:10.3390/cells13171467_

Round 1

Reviewer 1 Report

Comments and Suggestions for Authors

In this review, Yamazaki highlighted the regulatory role of IκBζ to NF-κB pathway, which is distinguished from other IκB family members. Unlike cytoplasmic IκB, IκBζ is special for its property by regulating NF-κB activity within the nucleus and its involvement in IL-17-triggered immune responses. Author offered a detailed overview from the structure, gene-regulation and cell-type specific roles of IκBζ. This review offers insight to the close connections between IκBζ and various inflammatory conditions, highlighting its role in both protective and pathological processes in human diseases. Due to the pivotal role of IκBζ in several cell types and association with human disease in section 4, reviewer suggested that author may supplement another table to summarize the current discovered inducer (like IL-33 mentioned in page 9) or inhibitor (like siRNA and dimethyl itaconate in page 7) targeting IκBζ and their related disease. A discussion of their potential applications in pre-clinical and clinical therapy would emphasize the importance of IκBζ in human disease and further improve the review.

Overall, the review provides a thorough study of the regulatory mechanisms of IκBζ and current studies involving its special role in different cell types. The review is well-organized and supplements with abundant details of IκBζ research. Therefore, the reviewer considers it suitable for acceptance by Cells.

Comments on the Quality of English Language

The English in this review is clear and technically sound. Only minor polishing is needed.

Reviewer 2 Report

Comments and Suggestions for Authors

                  This review manuscript provides a comprehensive overview of IκBζ, covering its first discovery, molecular functions and pathophysiological roles, with recent updates which connected to IL-17 and POU transcription factors. The texts are well-structured and clearly written, making it accessible to readers who may not familiar with this topic. Although some figures are too much simplistic (I have included specific comments in below), they are nonetheless informative and contribute to the overall understanding of the text. This review is certainly valuable and worthy of publication. However, I have a few minor comments and suggestions for improvements to enhance the clarity and quality of the manuscript.

Comments:

1.        Abstract: The author mentioned about human GWAS, but the texts seem only provided a limited discussion of these studies. Specifically, references 75 and 76 focus on for intestinal epithelial cells but do not extensively cover other cell types, with most studies derived from mouse models. Given the relevance to human diseases, it would be beneficial to include more detail on how human GWAS studies together with KO studies in mice contribute to our understanding of the pathophysiological role of IκBζ in section 4.

2.        Introduction: This manuscript could benefit from a clearer differentiation between the roles of cytoplasmic and nuclear IκB proteins, as this distinction is very important for understanding the unique function of IκBζ. While Table 1 summarizes these differences, an introductory paragrah that provide more detail comparison would enhance reader comprehension.

3.        L59-60: Related to the point above, it would be highly informative if authors include an alignment of amino acids, indicating where this specific residue is located in comparison to conventional cytoplasmic IκB proteins. This information could be integrated into Figure 1 for better visual representation.

4.        Figure 1: Because IκBζ  preferentially binds to p50 rather than p65, it would be more accurate to adjust the double arrow to be shifted onto p50 but not in between p50-p65 complex.

5.        Figure 3: Please clarify in the figure where exactly the primary and secondary response genes are located. If the red and blue arrows represent these responses, please indicate this in the figure legend. Additionally, please clarify the meaning of the gray arrows. Revisions to this figure would help general readers better understand the content.

6.        Figure 4: If possible, it would be beneficial to include representative marker genes or pathways associated with antimicrobial activity and pro-inflammatory mediators. For example, specifying genes or cytokines such as IL-17 and IL-36, which are mentioned in the texts, would provide more informative context.

7.        Table 2: It appears that the results of IκBζ KO in NK cell are not included in Table 2. Could you clarify why this is the case? Is it because the study did not use the Cre driver system?
